# Assessment of the Annual Erosion Rate along Three Hiking Trails in the Făgăraș Mountains, Romanian Carpathians, Using Dendrogeomorphological Approaches of Exposed Roots

Mihai Jula and Mircea Voiculescu *

Department of Geography, Faculty of Chemistry, Biology and Geography, West University of Timişoara, 4 Bd. Vasile Pârvan, 300223 Timişoara, Romania
* Correspondence: mircea.voiculescu@e-uvt.ro

**Abstract:** Mountain hiking trails are vital components of tourist infrastructure and provide recreational opportunities for a large number of tourists. Exposed roots along the tourist trails in the forested mountains are impacted by tourist trampling and various natural processes, thus becoming even more exposed and eroded. The aim of our study was to estimate the annual erosion rates along three hiking trails in the Făgăraș Mountains using dendrogeomorphological approaches. The three used routes were: Bâlea Hotel—Bâlea Waterfall (BWFHT), Bâlea Hotel—Bâlea Glacial Lake (BLHT), and Bâlea Hotel—Doamnei Glacial Valley (DVHT). The average annual erosion rates in BWFHT, BLHT, and DVHT were $10.6 \pm 4.4$, $6.8 \pm 3.9$, and $6.1 \pm 3.3$ mm·y$^{-1}$, respectively. Over a 56-year interval (1965–2021), 610 scars were recorded among the annual growth rings of the sampled tree roots; 172, 213, and 225 scars were recorded in BWFHT, BLHT, and DVHT, respectively. Moreover, we identified 1022 rows of traumatic resin ducts (TRDs) associated with scars: 237, 343, and 442 in BWFHT, BLHT, and DVHT, respectively. Additionally, the climate of the Făgăraș Mountains is humid with a multiannual average precipitation of 1366.2 mm; the precipitation in 24 h, between 1979 and 2021 in seven and three cases exceeded 70 mm/24 h and 100 mm/24 h, respectively. Thus, there were synchronous situations of root exposure with 24 h rainfall. However, it is unclear whether precipitation plays a decisive role in root exposure or in triggering erosion processes on tourist trails. We considered that tourist traffic plays a decisive role in root exposure and erosion, however locally and complementarily, 24 h precipitation must also be considered.

**Keywords:** hiking trails; dendrogeomorphological approach; exposed roots; scars; mean erosion rates; Făgăraş Mountains; Romanian Carpathians

## 1. Introduction

Tree roots can be exposed along hiking trails from forested areas at different angles as a result of both geomorphic processes and trampling and can be used in dendrogeomorphological studies to determine soil erosion rates [1–3]. The advantage of using exposed roots over other erosion assessment techniques is that erosion rates have an annual resolution [4]. Roots can be exposed and damaged by tourist trampling activities [5–7], wildlife such as deer [8] or caribou [9–12], natural processes such as extreme rainfall events [13,14], or geomorphic processes such as continuous denudation or erosive events [2,15]. Trampling activity causes bark removal, cambium exposure [16], and scar and traumatic resin duct formation near the wound as a typical response [3,17]. Scars can be used as indicators of the trampling activity on exposed roots [7,18,19]. TRDs are formed in response to a broad spectrum of external factors, such as insect and fungal attacks [20–25], extreme climatic conditions [26–30], and mechanical damage [17,31,32].

Exposed tree roots provide information on scars and radial growth changes, which can help in reconstructing geomorphological processes at the Earth's surface and estimate annual erosion rates [2,5,6,19,33–39].

The first investigations of erosion rates using exposed roots were conducted as early as the 1960s by Refs. [40–42] in the White Mountains of California (U.S.A.). Subsequently, in the 1970s, Ref. [43] highlighted the growth suppression and eccentric root development in the lower part of the roots.

In a study in the Piceance Basin, Colorado, Ref. [44] considered that the year of first root exposure was when the first scars appeared and the ring shape changed from concentric to eccentric. In the late 1990s, several interesting studies analysed the trampling scars on exposed roots to obtain extensive information on the movement of caribou (*Rangifer tarandus* L.) in northeastern Quebec-Labrador over the last 100 years, and in the Northwest Territories in Canada [10–12,45]. In the 2000s, studies were conducted to determine the first year of root exposure based on anatomical changes [2,4,46]. Subsequently, Ref. [47] assessed the erosion rates using exposed roots in the Alpes de Haute-Provence and demonstrated that root exposure results in structural changes when erosion reduces the soil cover to less than 3 cm. Therefore, they included a bias ($\varepsilon$) in the erosion rate calculation formula that was used in subsequent studies that focused on exposed roots to assess erosion rates [19,34,48–50].

As mentioned by Ref. [19], dendrogeomorphologists have also focused on areas with anthropogenic influence, such as tourist trails, especially in the last 20 years. Recently, there has been an increase in travel to forested mountain areas, national parks, and nature reserves owing to the interest and attractiveness of these areas, drawing a large number of tourists [51–54]. All types of mountain tourist activities are organised within a characteristic infrastructure comprising a network of marked and frequently used tourist trails [55–59] that prevent the uncontrolled dispersion of visitors [60,61].

The intense tourist activity in mountainous areas has different forms, most of which trample exposed roots during hiking. This triggers and intensifies soil erosion processes, causes damage to exposed roots, and degrades hiking trails, which is a worldwide problem [19,53,62]. The overuse of hiking trails threatens the natural and recreational value, of protected areas, and represents a real management challenge [63,64]. The most common indicators of trampling degradation are soil compaction and soil erosion, which are processes that are accelerated in steep terrain [65–68], trail widening and deepening [66,69,70], reduction or destruction of vegetation [71–73], and large-scale root exposure and damage [5–7,33,34,46]. Therefore, changes in root morphology and anatomy will underpin dendrogeomorphological approaches to establish a chronology of erosion events over time [1].

In Romania, except for a recent study [74], studies regarding mountain hiking trails and damage to roots through trampling and erosion are insufficient. Therefore, the aims of this study were to (i) identify the number and age of scars recorded in *Picea abies* (L.) H.Karst. roots produced by hiking along the three hiking trails in the Făgăraș Mountains; (ii) evaluate the annual erosion rates along these trails using a dendrogeomorphological approach; and (iii) assess the extent to which root ring reactions of trees located along the three mountain hiking trails can be used to highlight the intensity of tourist traffic.

## 2. Methodology

### 2.1. Study Area

Our study was conducted in the Făgăraș Mountains, located in the eastern part of the Southern Carpathians in the Romanian Carpathians at the intersection of the parallel of 45°30′ N and the 24°30′ E meridian (Figure 1).

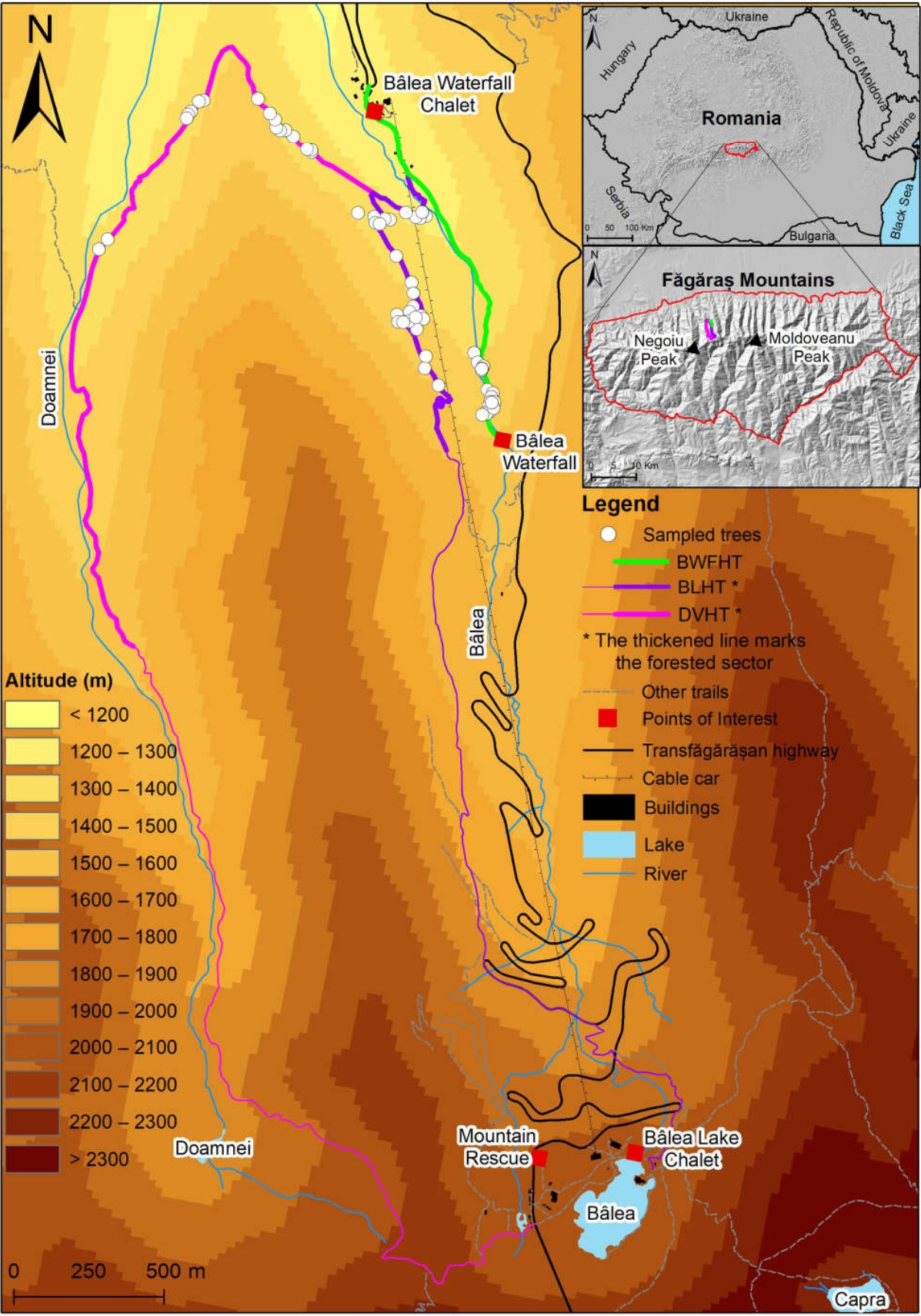

**Figure 1.** Geographical position of the Făgăraș Mountains and the studied tourist trails.

The Făgăraș Mountains cover an area of approximately 1500 km$^2$ and are shaped like a huge alpine ridge that is approximately 80 km long from East to West, from which two macro slopes detach into the northern and the southern ones [75]. The Făgăraș Mountains is the highest mountain area in Romania with altitudes exceeding 2500 m (Moldoveanu—2544 m a.s.l. and Negoiu 2535 m a.s.l.) and preserves landforms of Quaternary glaciations such as well-developed glacial cirques, U-shaped valleys, moraines, and erratic blocks [75]. The main periglacial landforms such as scree deposits, rock glaciers, solifluction, and snow avalanche tracks are also present. In this alpine ensemble, the predominant geomorphic processes were represented by erosional processes, snow avalanches, rockfall, rockslides, and debris flow. The general geology is crystalline and comprises chlorite, sericite, and kortitic systems, with intercalations of crystalline limestones and amphibolite [75].

The climate of the Făgăraș Mountains is harsh depending on geographic position and altitude. The northern slope is influenced by moist air from the Atlantic Ocean and cold airflows from the Arctic Ocean with low temperatures and the highest amount of snowfall. The southern slope is predominantly influenced by warm and moist air from the Mediterranean Sea with abundant snowfall and high snow depths. Therefore, according to the Koeppen-Geiger climate classification [76], the climate of these mountains is the Dw type (a snowy climate with dry winters). According to the seasonal snow cover classes [77], the Făgăraș Mountains climate shows the conditions of the alpine climate.

According to long-term weather records from the Bâlea Lake station (2044 m a.s.l., 45°36′ N, 24°37′ E; observation period 1979–2015), the mean annual air temperature (MAAT) is 0.2 °C, while the temperature in the winter season is −4.1 °C. The mean annual precipitation (MAP) was 1213.7 mm. The highest amount of precipitation occurs in the summer season (e.g., June—206.4 mm). Between 1 November and 31 May, precipitation mainly occurs as snow (661.7 mm w.e.). The mean number of days with snowfall exceeded 160 and the ground was covered with snow for 240–250 days per year.

The vegetation of the Făgăraș Mountains was differentiated according to the altitude [75]. Above the timberline (located between 1550 and 1900 m on the northern slope and between 1650 and 2000 m on the southern slope), the surfaces are covered by herbaceous vegetation grazed by sheep during the short alpine summer (3–4 months/year) and different shrub species (*Rhododendron kotschy* Simonkai, *Vaccinium myrtillus* L., *Vaccinium vitis idaea* L., *Pinus mugo* Turra, and *Juniperus nana* Willd.). At the highest elevations, lichens and mosses cover slopes. Under the timberline, dense Norway spruce (*Picea abies* L. Karst.) forests cover the slopes. Fir (*Abies alba* Mill.), pine (*Pinus sylvestris* L.), and larch (*Larix decidua* Mill., ssp. carpatica) trees are also found occasionally, whereas at lower altitudes, beech trees (*Fagus sylvatica ssp. sylvatica* L.), especially elm (*Ulmus glabra* Huds.) or maple (*Acer pseudoplatanus* L.), are predominant.

### 2.2. Hiking Trails

Three tourist trails located in the spruce forests of the Făgăraș Mountains (see Figure 1) were selected for the study: Hotel Bâlea-Bâlea waterfall (BWFHT), Bâlea Hotel-Bâlea Glacial Lake (BLHT), and Bâlea Hotel-Doamnei Glacial Valley (DVHT). The trails have different locations and dimensional characteristics. Knowledge of the relief microtopography is important because it can be a critical factor in assessing erosion rates using a dendrogeomorphological approach [19,34,78,79]. The alignments of the trails with respect to contour lines and the relationship between the trail route and existing landforms also play an important role in the evolution of the erosion process [80]. All three trails have a starting point at approximately 1240 m a.s.l.

BWFHT has a short route that is perpendicular to the contour line with two segments: the first is 410 m long, has a gentle slope and a width of about 2–2.5 m, while the second segment is steep with the occasional presence of stones, and is 957 m long and 1–1.5 m wide with exposed and trampled roots. Although the trail was badly damaged by a storm in 2020 that uprooted trees and broke stems, it had high tourist traffic, especially on weekends

and holidays. In various places, trees grow along the axis of the path and are bypassed on either side by tourists.

The BLHT has a constant relative width of approximately 1.5–1.8 m and a sideway alignment that intersects the slopes at different angles. For this reason, tourists create short-cuts. Trees and roots were situated on either side of the path and had obvious scars. Along the path, there are some maintenance facilities such as railings and wooden pavements.

DVHT detaches from BLHT and has a variable width between 1–1.5 m and a first sector that follows the contour lines and a second sector that is almost perpendicular to the contour lines. There are numerous trees with exposed roots located on either side of the path. Trees appear occasionally along the axis of the path, and the roots that are heavily trampled by tourists have obvious scars. The trail codes, their locations, and dimensional characteristics are presented in Table 1.

**Table 1.** General characteristics of the investigated trails (forested sector).

| Name/Code | Starting Point (m) | Arrival Point (m) | Vertical Drop (m) | Length (m) | Width (m) | Difficulty |
|---|---|---|---|---|---|---|
| Bâlea Hotel-Bâlea waterfall/BWHT | | 1430<br>45°36′10.199″ N<br>24°36′49.312″ E | 196 | 1367 | 1.2–1.5 | medium-high |
| Bâlea Hotel-Bâlea Lake/BLHT | 1234<br>45°38′10.903″ N<br>24°36′23.227″ E | 1560<br>45°37′31.4184″ N<br>24°36′37.6956″ E | 326 | 1972—forested | 1.5–2 | low-medium |
| Bâlea Hotel-Doamnei glacial valley/DVHT | | 1580<br>45°37′25.2264″ N<br>24°35′40.596″ E | 346 | 3767—forested | 1–1.5 | low-medium |

The geology of the three hiking trails comprised mica and paragneisses with intercalations of amphibolite, occasional crystalline limestones, and graphitic systems [81]. The predominant soil is acid brown. Podzolic clay loam soils are characteristic of contact with the subalpine floor [82].

The Romanian National Institute of Statistics does not record the number of tourists entering each tourist trail in the Făgăraș Mountains, similar to the situation in the Bucegi Mountains [74]. This is similar to the situation mentioned by Ref. [34] in their study of Ordesa and Monte Perdido National Park in the Central Spanish Pyrenees. Therefore, we do not have concrete data on tourist traffic.

### 2.3. Sampling Strategy

In the field, we identified exposed roots that were in contact with the soil and had visible trampling scars on their surface (Figure 2). Depending on the size of the roots and the surface area exposed to trampling, scars are easily recognisable due to their position on top of the roots and their round, elliptical, or elongated shape [10,45].

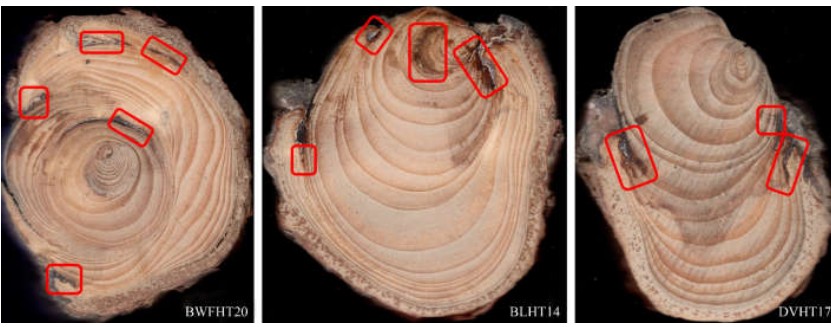

**Figure 2.** Cross sections of a spruce root. Rectangles indicate an example position of the identified trampling scars.

Sampling was conducted on three successive days in August 2021. Cross sections of *P. abies* roots with a thickness of 2–3 cm were obtained using a hand saw. We collected 20, 33, and 41 root samples from BWFHT, BLHT, and DVHT, respectively. We only sampled live roots, parallel to the direction of the flow. This orientation is indicated in the analysis of denudation values. Roots with other orientations can form small dams trapping the upstream sediments, which makes it difficult to obtain a reliable estimate of denudation [33,79].

Six samples were excluded because of the difficulty in recognising *P. abies* rings. Tree trunks were also sampled to determine their ages.

Depending on the width of the trails, we obtained samples about 1–1.5 m away from the stems of the trees [14,18,50]; this distance was sometimes larger than 1.5 m depending on the topography [33,34]. This distance was chosen for two reasons: to prevent disturbances caused by movements at the stem base that tends to push the roots upward and avoid root bending stress due to stem displacement [2,4,44,50].

Finally, using a Garmin GPS 76CSx, we recorded the geographical coordinates (longitude, latitude, and altitude) of each tree and root, which were then transposed onto a DEM with a resolution of 10 m. The distance between the root surface and the actual position of the soil layer was measured using a Burg Wachter Precise PS 7215 electronic calliper (accuracy of 0.01 mm), and the specific slope of each root was measured using an electronic clinometer (error of 0.25°).

### 2.4. Sample Analysis

Samples were prepared in the laboratory and air-dried for 2 months [33,34]. To facilitate the recognition of the growth rings of trees, the samples were progressively polished using sandpapers of different grits (400, 600, 800, and 1000) [12,44,49]. The analyses were conducted according to standard dendrogeomorphological procedures [1,4,7,19]. To this end, samples were scanned at a resolution of 1200 dpi to highlight the annual growth rings, scars, callus tissue, and TRDs.

The range of the analysis was considerably long (56 years). Therefore, a reference chronology was constructed to accurately estimate the age of roots. We counted and measured the thickness of the rings on four radii of the root cross section to minimise uncertainties due to false or missing rings [4,6,83,84] and highlight the variability of the width of the growth ring [79]. For this procedure, we used CooRecorder 9.0 and CDendro 9.0 software. The age of the exposed roots was determined by counting the annual growth rings [7,33,50].

### 2.5. Characteristics of Trampling Scar and Scar Dating and TRD

The scars in our study area were formed as a result of the trampling activity of hikers, especially between June and October. We did not find any traces of fire on the scars, such as charcoal or frostbite scars, which are easily distinguishable from trampling activity scars [12].

Scarring occurs from a trampling process that causes cambium death in the upper part of the roots and stops radial growth in that root section [8,10–12,46]. Trampling scars are easy to recognise based on the root thickness, position on the roots, and shape [10–12]. The size of the scars was influenced by the root diameter. Scars were located on the upper side of the exposed roots and on both sides of the roots. Occasionally, they can be located in the same position as overlapping scars [12]. Scars are formed at the boundary between the exposed and unexposed root rings, which indicates the time of first root exposure [44]. The age of a scar can be determined by counting the annual rings added after lesion formation. As some rings may be absent from the samples, cross-dating is essential [10,85]. The exact year of scar formation was only considered when the lesion was produced during xylem growth. Even under cross-dating conditions, there may still be acceptable errors of +/−1 year, possibly owing to the seasonality of the inactive cambial phase. Therefore, the dating spans two calendar years: the autumn of one year to the spring of the next year [10].

TRD is an anatomical feature of wood that is associated with root lesions [31]. They usually form within a tree ring produced in the year of injury, immediately after the event, but can also form later in several consecutive years [3,17], especially if trees are injured during the dormancy period preceding the growth season [86]. In the case of hiking trails, the mechanical impact of trampling can lead to the removal of the cambium and the formation of scars and TRD in proximity to the wound as a typical response; but this response is not immediate. As Ref. [17] argues: 'TRD constitutes the most commonly observed signature of a past growth anomaly on increment cores and some 44–86% of reconstructed growth disturbances would have remained undetected, had the presence of TRD not been considered'.

Thus, to highlight the following: (a) the trampling impact of tourist activities over the exposed roots situated along hiking trails, and (b) the continuous and intense use of the trails by hikers in the absence of data on the actual number of tourists provided by the authorities, TRDs were assessed and an index was assigned to each growth ring as follows: 0 = no impact, 1 = very reduced impact (<10 ducts), 2 = reduced impact (between 10 and 30 ducts), 3 = medium impact (>30 ducts are present, but less than a quarter of the ring is affected), 4 = strong impact (between a quarter and half of the ring is affected), and 5 = very strong impact (>half of the ring is affected). The annual impact (Ia) was then calculated for each year by summing the indices from each root using the following normalized equation:

$$Ia(t) = [(n_0(t) \times 0) + (n_1(t) \times 1) + (n_2(t) \times 2) + (n_3(t) \times 3) + (n_4(t) \times 4) + (n_5(t) \times 5)]/n(t)$$

where t = the considered year and n = number of rings with the specific index

### 2.6. Estimation of the Erosion Rate

There is a relationship between the distance of exposed roots from the soil and the number of years since their emergence [1,7,87]. The following parameters are required to obtain the erosion rates ($E_{ra}$) [47]: the number of rings since the time of exposure ($NR_{ex}$) and the thickness of the eroded soil layer after exposure ($E_x$). The overestimation of $E_{ra}$ was corrected by considering secondary root growth in the upper and lower parts of the root. While erosion processes cause continuous and progressive denudation, some anatomical parameters undergo changes even before the root is exposed [47] (e.g., the tracheid lumen starts to decrease when the soil cover is less than a few centimetres thick). Thus, the assessment of the mean annual erosion rates would be underestimated if this bias ($\varepsilon$) was not considered [47]. The use of ImageJ 1.52u and CooRecorder 9.0 enabled the measurement of root ring width and the identification of growth anomalies [88]. We calculated erosion rates using the formula given by Ref. [47], which has recently been used in other studies.

$$Era \;=\; \frac{E_x - (G_{r1} - G_{r2}) + \left(\frac{B_1 + B_2}{2}\right) + \varepsilon}{NR_{ex}} \tag{1}$$

where $E_{ra}$ = annual erosion rate (mm·y$^{-1}$); $E_x$ = height of the exposed part of the root (mm); $G_{r1}$ and $G_{r2}$ = secondary growth, after exposure, in the upper and lower parts of the root (mm); $B_1$ and $B_2$ = bark thickness in the upper and lower parts of the root (mm); $\varepsilon$ = bias = 30 (mm); $NR_{ex}$ = number of annual rings formed after the time of root exposure.

Mean, standard deviation (SD), minimum (min), and maximum (max) were used to describe the data. To evaluate the statistical significance of different values of the erosion rate between roots with exposures up to 20 years and those with exposures over 20 years, the Mann-Kendall test was conducted with a statistical significance threshold of $p < 0.05$.

## 3. Results

### 3.1. Root Characteristics

The mean age of the roots is approximately similar: 31.8 ± 15.4, 34.1 ± 16, and 39 ± 11.9 years in BWFHT, BLHT, and DVHT, respectively. The youngest and oldest roots

in BWFHT were 10 and 76 years old, respectively; in BLHT, the youngest and oldest roots were 18 and 74 years old, respectively; and in DVHT, the youngest and oldest roots were 19 and 66 years old, respectively. The average root diameter in BWFHT was 56.3 ± 8.2 mm, while those in the other two hiking trails were close: 47.9 ± 10.2 and 45.2 ± 8.2 mm in BLHT and DVHT, respectively. The characteristics of the exposed roots from which the samples were collected are presented in Table 2.

**Table 2.** Characteristics of the exposed roots (*Picea abies*) and measurements of parameters assessed along BWFHT, BLHT, and DVHT.

| Sample Code | Root Age (years) | First Year of Exposure | Age of Exposure | Diameter (mm) | $E_x$ (mm) | $E_{ra}$ (mm·y$^{-1}$) |
|---|---|---|---|---|---|---|
| BWFHT1 | 18 | 2013 | 8 | 47.88 | 20 | 9.41 |
| BWFHT2 | 22 | 2008 | 13 | 65.76 | 95 | 12.83 |
| BWFHT3 | 24 | 2007 | 14 | 65.73 | 80 | 11.25 |
| BWFHT4 | 29 | 1999 | 22 | 57.37 | 90 | 7.41 |
| BWFHT5 | 37 | 1994 | 27 | 50.55 | 120 | 6.24 |
| BWFHT7 | 48 | 1987 | 34 | 54.81 | 110 | 4.40 |
| BWFHT8 | 26 | 2008 | 13 | 53.44 | 100 | 12.64 |
| BWFHT9 | 76 | 2009 | 12 | 52.47 | 110 | 13.42 |
| BWFHT10 | 55 | 2004 | 17 | 53.86 | 200 | 15.04 |
| BWFHT11 | 10 | 2016 | 5 | 47.51 | 70 | 24.04 |
| BWFHT12 | 31 | 1999 | 22 | 61.98 | 90 | 7.45 |
| BWFHT13 | 23 | 2004 | 17 | 62.71 | 90 | 8.97 |
| BWFHT14 | 32 | 1998 | 23 | 68.1 | 175 | 10.21 |
| BWFHT15 | 26 | 2008 | 13 | 49.44 | 60 | 9.33 |
| BWFHT16 | 16 | 2011 | 10 | 42.63 | 120 | 16.66 |
| BWFHT17 | 34 | 2001 | 20 | 59.23 | 120 | 9.18 |
| BWFHT18 | 19 | 2009 | 12 | 50.94 | 110 | 14.21 |
| BWFHT19 | 31 | 2000 | 21 | 45.71 | 75 | 5.96 |
| BWFHT20 | 55 | 1990 | 31 | 75.37 | 130 | 5.42 |
| BWFHT21 | 25 | 2006 | 15 | 60.63 | 80 | 9.42 |
| **Mean** | **31.8 ± 15.4** | - | **17.5 ± 7** | **56.3 ± 8.2** | **102.3 ± 37.8** | **10.7 ± 4.5** |
| BLHT1 | 30 | 2010 | 11 | 53.11 | 140 | 16.12 |
| BLHT2 | 25 | 2006 | 15 | 46.71 | 70 | 8.40 |
| BLHT3 | 67 | 1990 | 31 | 61.43 | 110 | 4.89 |
| BLHT5 | 21 | 2010 | 11 | 41.27 | 60 | 8.86 |
| BLHT6 | 38 | 1998 | 23 | 41.02 | 50 | 4.24 |
| BLHT7 | 31 | 2005 | 16 | 41.04 | 30 | 4.68 |
| BLHT8 | 18 | 2010 | 11 | 47.32 | 60 | 11.14 |
| BLHT9 | 13 | 2014 | 7 | 38.28 | 20 | 10.32 |
| BLHT10 | 24 | 2012 | 9 | 30.75 | 60 | 11.97 |
| BLHT12 | 55 | 1978 | 43 | 54.79 | 60 | 2.55 |
| BLHT14 | 25 | 2004 | 17 | 52.14 | 40 | 6.18 |

**Table 2.** *Cont.*

| Sample Code | Root Age (years) | First Year of Exposure | Age of Exposure | Diameter (mm) | $E_x$ (mm) | $E_{ra}$ (mm·y$^{-1}$) |
|---|---|---|---|---|---|---|
| BLHT16 | 41 | 1994 | 27 | 59.82 | 100 | 5.70 |
| BLHT17 | 21 | 2005 | 16 | 42.6 | 35 | 5.38 |
| BLHT18 | 28 | 2000 | 21 | 45.2 | 30 | 4.19 |
| BLHT19 | 30 | 1999 | 22 | 55.18 | 140 | 9.48 |
| BLHT20 | 64 | 1987 | 34 | 52.24 | 40 | 2.36 |
| BLHT21 | 19 | 2012 | 9 | 41.13 | 90 | 15.32 |
| BLHT22 | 49 | 1985 | 36 | 38.67 | 20 | 1.89 |
| BLHT23 | 60 | 1985 | 36 | 42.13 | 40 | 2.67 |
| BLHT24 | 26 | 2007 | 14 | 42.01 | 70 | 9.34 |
| BLHT25 | 17 | 2013 | 8 | 46.58 | 40 | 12.22 |
| BLHT26 | 44 | 1997 | 24 | 43.26 | 20 | 2.93 |
| BLHT27 | 27 | 2001 | 20 | 62.19 | 60 | 6.30 |
| BLHT28 | 74 | 1959 | 62 | 44.68 | 50 | 1.71 |
| BLHT29 | 61 | 1970 | 51 | 64.89 | 40 | 2.09 |
| BLHT30 | 18 | 2012 | 9 | 43.34 | 40 | 10.35 |
| BLHT31 | 27 | 2003 | 18 | 32.3 | 30 | 3.59 |
| BLHT32 | 35 | 2013 | 8 | 46.56 | 50 | 12.01 |
| BLHT33 | 25 | 2006 | 15 | 79.89 | 60 | 6.67 |
| BLHT35 | 41 | 1990 | 31 | 41.39 | 40 | 2.42 |
| BLHT36 | 24 | 2004 | 17 | 41.26 | 60 | 6.04 |
| BLHT37 | 25 | 2000 | 21 | 44.37 | 40 | 4.04 |
| BLHT39 | 24 | 2005 | 16 | 64.63 | 80 | 8.27 |
| **Mean** | **34.2 ± 16** | **-** | **21.5 ± 12.8** | **47.9 ± 10.2** | **56.8 ± 29.9** | **6.8 ± 3.9** |
| DVHT1 | 22 | 2005 | 16 | 38.45 | 100 | 9.52 |
| DVHT2 | 31 | 2003 | 18 | 36.58 | 25 | 3.49 |
| DVHT3 | 38 | 1994 | 27 | 40.04 | 70 | 4.52 |
| DVHT4 | 46 | 1995 | 26 | 51.24 | 30 | 2.87 |
| DVHT5 | 32 | 2006 | 15 | 36.83 | 30 | 5.18 |
| DVHT6 | 41 | 1987 | 34 | 45.14 | 140 | 5.40 |
| DVHT7 | 59 | 1994 | 27 | 40.01 | 100 | 5.38 |
| DVHT8 | 26 | 2011 | 10 | 40.18 | 50 | 9.41 |
| DVHT9 | 29 | 2001 | 20 | 45.96 | 80 | 6.35 |
| DVHT10 | 41 | 2003 | 18 | 57.68 | 160 | 12.33 |
| DVHT11 | 21 | 2010 | 11 | 24.7 | 70 | 10.09 |
| DVHT12 | 26 | 2002 | 19 | 44.77 | 100 | 8.38 |
| DVHT13 | 66 | 1984 | 37 | 57.29 | 80 | 3.41 |
| DVHT14 | 22 | 2007 | 14 | 49.58 | 60 | 8.43 |
| DVHT15 | 42 | 1986 | 35 | 66.16 | 60 | 3.77 |

**Table 2.** *Cont.*

| Sample Code | Root Age (years) | First Year of Exposure | Age of Exposure | Diameter (mm) | $E_x$ (mm) | $E_{ra}$ (mm·y$^{-1}$) |
|---|---|---|---|---|---|---|
| DVHT17 | 19 | 2014 | 7 | 40.21 | 50 | 13.98 |
| DVHT18 | 43 | 1996 | 25 | 39.4 | 50 | 3.48 |
| DVHT19 | 36 | 1993 | 28 | 41.55 | 80 | 4.60 |
| DVHT20 | 34 | 2004 | 17 | 41.64 | 40 | 4.84 |
| DVHT21 | 43 | 1988 | 33 | 53.73 | 80 | 3.94 |
| DVHT22 | 42 | 1990 | 31 | 38.67 | 70 | 3.73 |
| DVHT23 | 41 | 1994 | 27 | 50.04 | 60 | 3.98 |
| DVHT24 | 56 | 1990 | 31 | 52.18 | 90 | 4.87 |
| DVHT25 | 52 | 1991 | 30 | 46.55 | 70 | 4.15 |
| DVHT26 | 45 | 2000 | 21 | 40.51 | 70 | 5.56 |
| DVHT28 | 58 | 1991 | 30 | 40.03 | 55 | 3.08 |
| DVHT29 | 56 | 1994 | 27 | 40.04 | 80 | 4.54 |
| DVHT30 | 37 | 2005 | 16 | 57.58 | 70 | 6.81 |
| DVHT31 | 30 | 2004 | 17 | 43.78 | 30 | 4.07 |
| DVHT32 | 64 | 1973 | 48 | 46.82 | 50 | 2.14 |
| DVHT33 | 37 | 1999 | 22 | 49.49 | 140 | 8.59 |
| VDHT35 | 52 | 1982 | 39 | 55.97 | 100 | 3.91 |
| DVHT36 | 30 | 2004 | 17 | 49.67 | 60 | 7.15 |
| DVHT37 | 28 | 2002 | 19 | 65.6 | 110 | 9.75 |
| DVHT38 | 23 | 2015 | 6 | 39.14 | 70 | 19.41 |
| DVHT39 | 40 | 1992 | 29 | 46.3 | 50 | 4.05 |
| DVHT40 | 43 | 1999 | 22 | 32.71 | 50 | 4.12 |
| DVHT41 | 36 | 1995 | 26 | 41.77 | 90 | 5.49 |
| DVHT42 | 32 | 1998 | 23 | 44.39 | 40 | 3.60 |
| DVHT43 | 51 | 1991 | 30 | 39.91 | 60 | 3.48 |
| DVHT44 | 33 | 1997 | 24 | 43 | 120 | 7.46 |
| **Mean** | **39.1 ± 11.9** | **-** | **23.7 ± 8.7** | **45.3 ± 8.2** | **72.9 ± 30** | **6.1 ± 3.4** |

*3.2. Erosion Rate and Root Exposure*

The variation in erosion rates across the three trails was influenced by the effect of tourist trampling activity, root age, and year of exposure.

Within BWFHT, the erosion rates are between 4.40 and 24 mm·y$^{-1}$, with an average of 10.6 ± 4.4 mm·y$^{-1}$. One case had an erosion rate below 5 mm·y$^{-1}$, ten cases (47.6% of the total) had erosion rates between 5 and 10 mm·y$^{-1}$, eight cases (38.1%) between 10 and 20 mm·y$^{-1}$, and one case above 20 mm·y$^{-1}$. Within the BLHT, 25 (75.8%) and eight (24.2%) sampled roots showed average annual erosion rates below 10 mm·y$^{-1}$ and between 10 and 20 mm·y$^{-1}$, respectively. BLHT has a different situation, wherein the erosion rates were between 1.9 and 16.2 mm·y$^{-1}$, with an average of 6.8 ± 3.9 mm·y$^{-1}$. Of the total number of samples, 14 samples (42.4%) had an erosion rate between 1 and 5 mm·y$^{-1}$, 11 samples (33.4%) between 5 and 10 mm·y$^{-1}$, and eight (24.2%) samples higher than 10 mm·y$^{-1}$. A similar situation was also observed in the DVHT wherein the erosion rate was between 2.14 and 19.4 mm·y$^{-1}$, with an average of 6.1 ± 3.3 mm·y$^{-1}$. More than half of the samples (*n* = 22), representing 53.6% of the total, had an erosion rate between 1 and 5 mm·y$^{-1}$,

12 samples (29.2% of the total) between 5 and 10 mm·y$^{-1}$, and four samples (9.7%) higher than 10 mm·y$^{-1}$ (Figure 3).

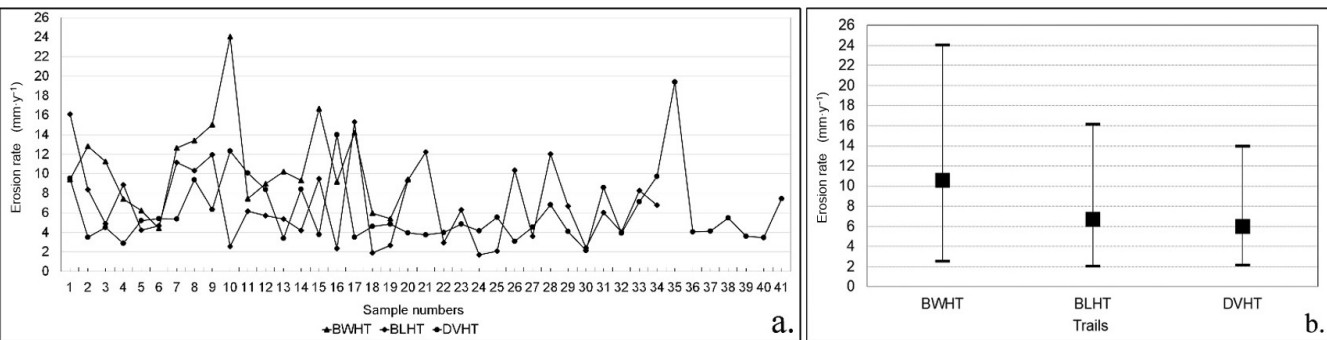

**Figure 3.** Erosion rates along the three trails (**a**) and maximum, average, and minimum erosion rates within each trail (**b**).

The sampled roots had different exposures; those with exposures between 10 and 20 years had the highest erosion rates. Long exposure of roots leads to underestimation of erosion rates (Figure 4).

In BWFHT, we identified 15 exposure events; the oldest and youngest exposures were dated to 1987 and 2016, respectively. The highest number of exposed roots occurred between 1998 and 2001 (*n* = 4, 26.6% of the total) and between 2003 and 2011 (*n* = 7, 46.6% of the total). Sixty percent of roots had exposures of less than 20 years, and the erosion rates were 1.9 times higher than those of roots with exposures of more than 20 years (12.8 ± 4 mm·y$^{-1}$/6.7 ± 1.7 mm·y$^{-1}$; *p* < 0.0152). The situation was different for the other two hiking trails. Twenty exposure events were recorded in BLHT; the oldest and most recent events were dated to 1970 and 2014, respectively. The highest number of exposed roots (*n* = 10; 50% of the total) was recorded between 1997 and 2007. Roots with exposures below 20 years represent 57.5% of the total trail and their erosion rates were 2.5 times higher than those of roots with exposures above 20 years (9.1 ± 3.3 mm·y$^{-1}$/3.6 ± 1.9 mm·y$^{-1}$; *p* < 0.0001). In DVHT, 29 exposure events were recorded; the oldest and most recent events were recorded in 1973 and 2016, respectively. The majority of the exposed roots (*n* = 18; 62% of the total) were recorded between 1990 and 2007. Roots with exposures up to 20 years represented 43.9% of the total trail and their erosion rates were 2.1 times higher than those with exposures above 20 years (8.8 ± 4.1 mm·y$^{-1}$/4.2 ± 1.1 mm·y$^{-1}$; *p* < 0.0002).

*3.3. Scars and TRD Distribution*

The intense impact of trampling due to hiking has been documented in the large number of scars and TRD in spruce roots exposed to the three hiking trails. A total of 610 scars were recorded between 1965 and 2021 (over a 56-year period). The mean number of scars was 9.66, 8.59, and 5.74 in BWFHT, BLHT, and DVHT, respectively. In BWFHT, BLHT, and DVHT, the oldest scars were recorded in 1987, 1965, and 1984, respectively. Their distribution could be followed within four intervals. From 1965 to 1984, scars were recorded only in the BLHT group. From 1984 to 1987, scars were also recorded in BWFHT and DVHT. Between 1988 and 1994, scars were recorded only in BWFHT and DVHT; thereafter, until the end of the observation period, scars were consistently present in all three hiking trails. Between 1995 and 2003, the number of scars increased (17.3% of the total). Between 2004 and 2018, with an increase in interest in tourist activities, the number of scars increased more, reaching a total of 460 (75.4% of the total). After this interval, their numbers started to decrease significantly, especially in 2020 and 2021, largely due to the coronavirus pandemic (COVID-19), which imposed accommodation and travel restrictions on tourists. The peaks in the number of scars occurred between 2016 and 2018 in BWFHT, in 2011, 2013, 2014, 2015, and 2018 in BLHT, and 2011 and 2015 in DVHT (Figure 5). An average of 13.21 rings showing TRD was registered in the BWFHT, the first ring with TRD being registered in

1988. In BLHT, an average of 17.33 rings showing TRD were recorded, and the first ring showing TRD was identified in 1963. In DVHT, the average number of rings showing TRD was 14.62, and the first ring with TRD was identified in 1985. In BWFHT, between 2012 and 2019, an average of six cases of TRD were recorded (62% of the total trail), in BLHT an average of 5.7 cases were recorded (28.8% of the total trail) between 2010 and 2018, and in DVHT an average of 12.45 cases were recorded (88.4% of the total trail) between 1996 and 2020 (Figure 5).

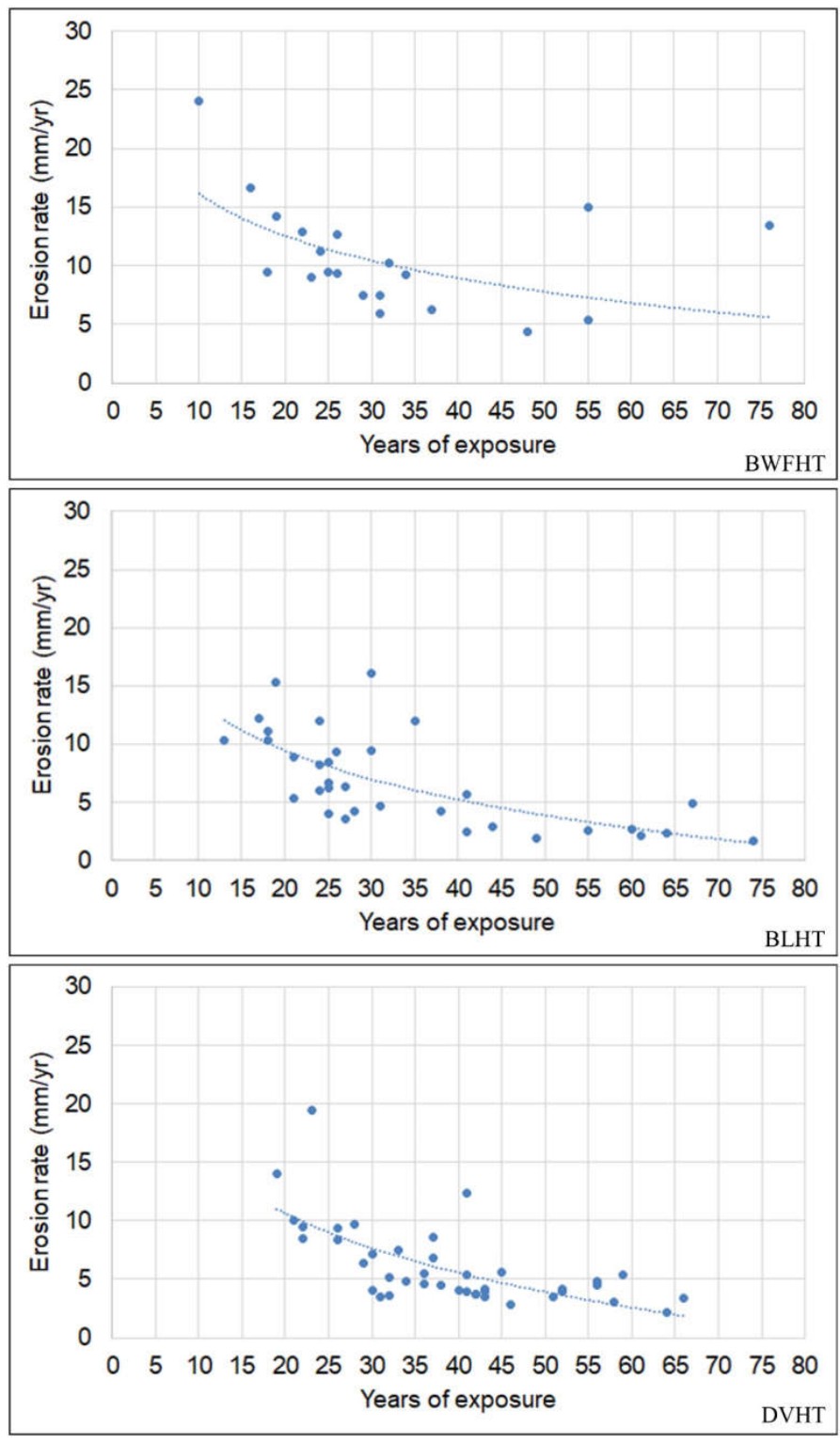

**Figure 4.** Erosion rates and the number of years of root exposure.

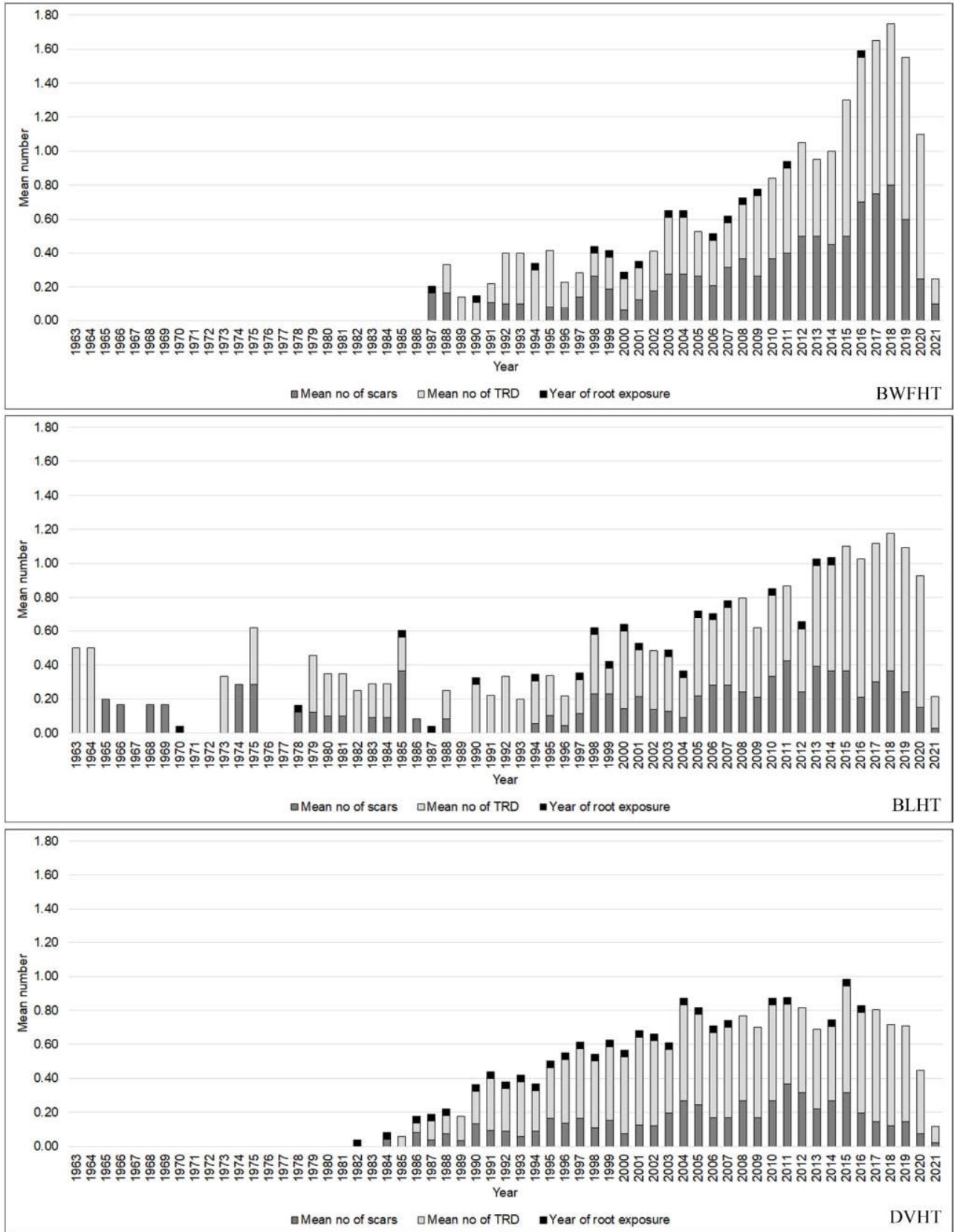

**Figure 5.** Number of scars and TRDs along BWHT, BLHT, and DVHT between 1963 and 2021.

Root exposure was intensified by the trampling activity, which was documented in a large number of TRDs identified in the exposure year, and consecutive years. Thus, in

BWFHT, we found TRDs in all the analysed roots, in BLHT and DVHT, 75.7 and 72.7%, respectively. Our analysis confirmed that exposed roots located along hiking trails that were continuously subjected to trampling have most of their TRDs located near the scars, almost as a continuation of the latter. The annual impact of trampling (Ia), using valid living samples, is shown in Figure 6.

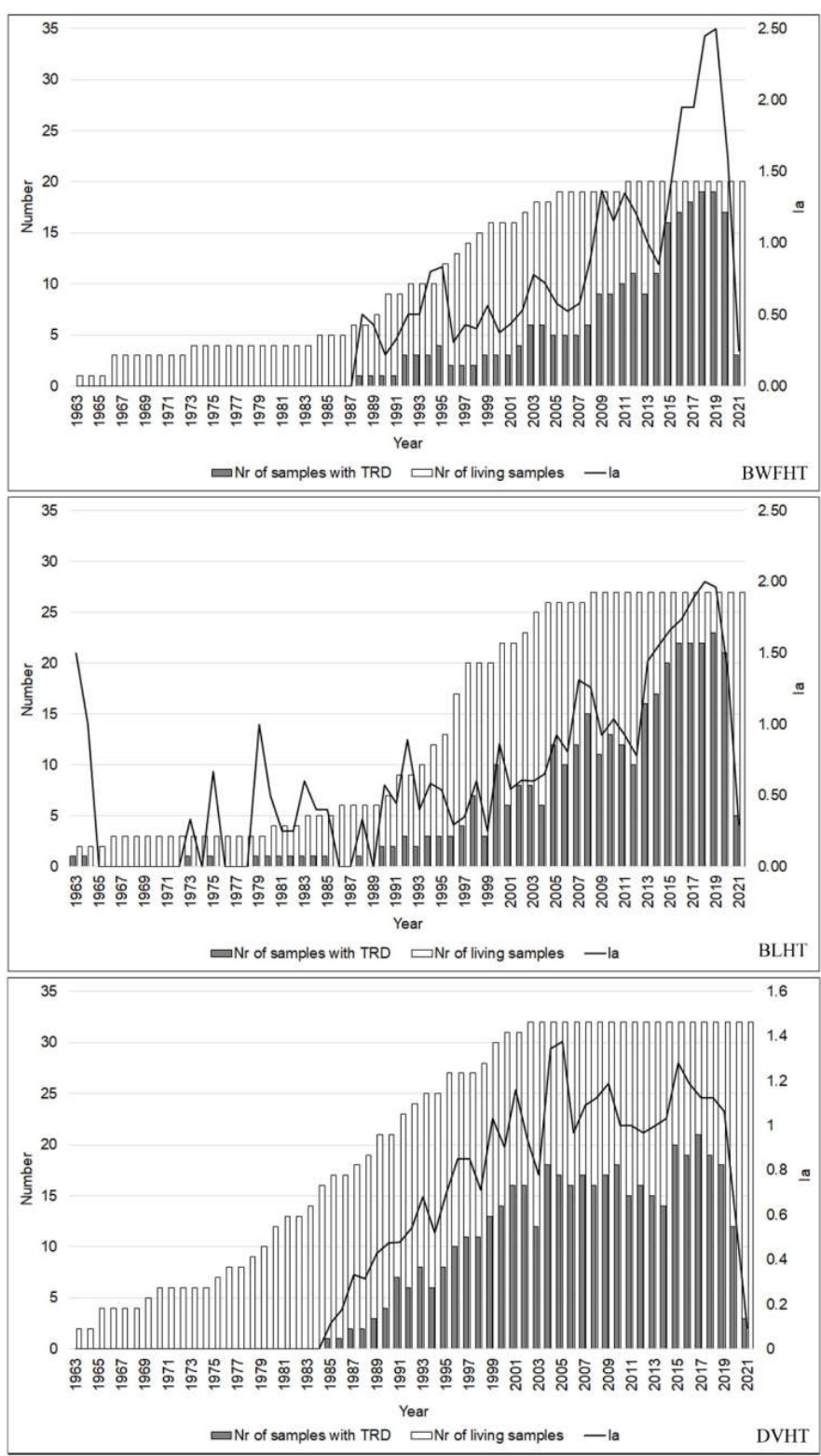

**Figure 6.** Annual impact along the three hiking trails.

## 4. Discussions

The dendrogeomorphological method proved to be a useful tool for highlighting and evaluating the scars recorded on the exposed roots along hiking trails and the annual erosion rate. Regardless of the intensity of use, the conifer roots retained their ability to form scars [12].

The number of scars and the average annual rate of erosion must be understood within the context of the economic and tourism transformations that have occurred in Romania since the revolution in December 1989.

BWFHT is short and heavily used only as an access conduit to the Bâlea waterfall, which takes approximately 40–50 min. The area of this trail is mainly a destination (camping and caravanning) for weekend and holiday tourism, which attracts many tourists. The number of scars increased significantly since 2001, and then from 2012 to 2019.

BLHT and DVHT, with lengths of 1972 and 3767 m, respectively, facilitate the entrance of tourists into the high alpine area of the Făgăraș Mountains. These two hiking trails have serious competition in the movement of tourists due to the Bâlea cable car and the Transfăgărășan highway. The Bâlea cable car, which is 3700 m long, was built in the early 1970s above the Bâlea glacial valley between the minimum and maximum final altitudes of 1200 and 2040 m, respectively. The Transfăgărășan highway was built between 1970 and 1974 with a series of improvements until 1980. From the intersection with the trailhead at the Bâlea Hotel to the Bâlea glacial lake, this route is approximately 14 km long. This means of transport contributes to the movement of a good number of tourists who want to reach the high part of the mountains. Notably, BLHT is one of the oldest tourist trails in the Făgăraș Mountains. In BLHT, between 1970 and 1973, no scar was recorded because of the work on the Transfăgărășan highway when the Bâlea glacial valley was closed. In 1974, the Transfăgărășan highway was inaugurated and BLHT opened to tourist traffic.

Between 1974 and 1988, the number of scars was reduced by 1–2 scars/year, except for the year 1985, with four scars. A high number of scars were recorded after 1994, with an increase in mountain tourism stimulated by private initiatives in the Bâlea glacial cirque area located at 2000 m a.s.l., such as the appearance of private huts, camping areas, and commercial activities. The Bâlea glacial cirque is the main place of entry to the central glacial sector of the Făgăraș Mountains and to the highest peaks above 2400–2500 m a.s.l. The DVHT is a parallel, longer version of the BLHT that leads to the high alpine area of the Făgăraș Mountains and is aimed at well-equipped and trained tourists. The scars have appeared since the mid-1980s, with a large increase between 2004 and 2015. Additionally, the actual number of hikers remains unknown, and there are no records of them either in the statistics of the tourist organisations in the area or at the entry points to the mountain. Therefore, we assumed that the numbers of trampling scars and erosion rates are determined by tourist traffic.

In the case of the BWFHT and BLHT, the small number of valid samples before 1990 makes it difficult to accurately assess the Ia index. In the case of the DVHT, the increase in trampling by tourists and its impact on roots is clear. In all three cases, a sharp decrease in Ia was observed in 2020 and 2021, which could be associated with a reduced number of tourists due to the COVID-19 pandemic. Notably, all samples were collected in August 2021; however, although this date corresponds to the middle of the tourist season, the downward trend for that year remains clear.

Erosion rates are close to those recently recorded in the Bucegi Mountains along two hiking trails under similar environmental conditions [74]: 6.8 mm·y$^{-1}$ in BLHT and 6.1 mm·y$^{-1}$ in DVHT, compared to 6.4 mm·y$^{-1}$ in SHT (Bucegi Mountains). The erosion rate of 10.6 mm·y$^{-1}$ recorded in the BWFHT is close to 14.1 mm·y$^{-1}$ recorded in the UHT in the Bucegi Mountains. The last two hiking trails have similar characteristics, as they are short and intensely used by hikers to reach waterfalls and are accessible to all hikers. Additionally, the erosion rates in our study are higher than those obtained by Ref. [7] in the Central Italian Alps (between 2.7 and 3.7 mm·y$^{-1}$) but lower than those obtained by Ref. [60] in two protected areas in south-central Poland (16 ± 25 mm·y$^{-1}$) or those

obtained in a national park in northern Japan (110 mm·y$^{-1}$; [89]). Erosion rates obtained in BLHT and DVHT are between those obtained in the Spanish Pyrenees (3.1 ± 1.5 and 8.9 ± 4.3 mm·y$^{-1}$, [34]).

The orientation of the trail to the prevailing slope, known as the trail angle [90] or slope alignment angle [91], plays an important role in the analysis of trails and in assessing root exposure and erosion. Thus, the importance of the slope alignment angle increased as the slope of the trail increased. Trails parallel to the slope under steep conditions are more prone to erosion, trampling, and slippage by tourists [92]. Thus, in BWFHT, with an average slope of 38.2° and a predominantly slope-parallel alignment, deep erosion occurred mostly between the 30° and 50° slopes. Additionally, as mentioned by Ref. [93], the steeper the slope, the poorer the maintenance of the trails; this also applies to the BWFHT. In the other two trails, we also mentioned that trampling contributed to root damage by hikers, as mentioned by Ref. [92]. In BLHT, sideways slope alignment angles alternate with parallel ones [80]; the average slope is 20° and higher erosion rates are concentrated in the range of 15° to 30°. In DVHT, the first part is a mix of lateral and perpendicular alignment, then, it is slope parallel. Here, the average slope was 24.6°, and the erosion rates were concentrated in two ranges: between the slope angles of 15° and 30°, and 30° and 40° (Figure 7). These values confirm that erosion increases with the slope and local topography, which are the most serious driving forces of environmental degradation [50,94,95]. These results are similar to those found by Ref. [49] and can be attributed to the intense trampling activity under mountain slope conditions.

False or missing rings are specific, not only to tree stems, but also to tree roots [96], and can affect the actual number of years that have elapsed since the onset of erosion [97]. Cross dating is used to eliminate the interpretation errors, and root cross-sectional analysis can minimise the resulting errors [14,84,88,98,99]. Root exposure and erosion can be amplified by the simultaneous action of natural geomorphological processes and human trampling. Our study area was located in a humid climate similar to that of the Western Carpathians, Southern Poland [1,14], or the Carpathians and Sudetes mountain ranges of the Czech Republic [100,101]. The multiyear average rainfall was 1366.2 mm. Therefore, we considered that the false rings are attributed to the presence of rocks in the soil that appear on the surface of the trails [33,102]. This is especially true for BWFHT, where a number of rocks appeared in the trail in its median part, and for BLHT and DVHT, where rocks appeared in the first sector and in a few other short sectors (Figure 8).

Precipitation, as a natural process, through its amount, duration, and intensity, can trigger and amplify erosion processes [70,103,104] under conditions of high slopes [1] and can induce root exposure [14]. There may be a certain link between high precipitation and erosion processes in a temperate climate, where our study area is located [83,105,106]. Therefore, in addition to trampling, the processes of root exposure and erosion may also be triggered by intense precipitation, which occurs in the area of the three hiking trails.

According to the data recorded at the Bâlea weather station, there were some years with high annual rainfall averages that exceeded the multi-year average of 1366.2 mm, reaching values of over 1600 mm in 1980, 1700 mm in 1981, 1800 mm in 2010, and 2100 mm in 2016 (Figure 9a).

However, we considered that root exposure and erosion rates are not controlled by precipitation, as we found no trend in the time series between 1979 and 2021 ($R^2$ = 0.2141), which was similar to that found by Ref. [34] in their study of hiking trails in the protected natural area in the Spanish Pyrenees. In our case, only a few root exposure and erosion events coincided with the occurrence of heavy rainfall events, which were synchronous. Thus, in BWFHT, only one root was exposed in the year with exceptional rainfall (2016; annual rainfall value was 2131.3 mm, which is 1.5 times higher than the annual average); in BLHT, three roots were exposed in 2010 (1804.5 mm of annual rainfall, which is 1.3 times higher than the annual average); and in DVHT, only one root was exposed in 2010. In contrast, more roots had asynchronous exposure, with fewer rainy years. Additionally, a number of roots were exposed in years with low rainfall amounts, below the annual

mean of the observation series of 1366.2 mm, owing to asynchronous conditions with high rainfall amounts. Thus, in BWFHT, one root was exposed in 1998, 2000, 2001, and 2006, and two roots in 1999 and 2004, representing 30% of the total roots sampled. In BLHT, one root was exposed in 1994, 1997, 1998, 1999, and 2003, and two roots in 1985, 1990, 2000, 2004, and 2006, representing 30.3% of the total roots in this trail. In DVHT, one root was exposed in 1982, 1986, 1992, 1993, 1996, 1997, 1998, 2000, 2001, 2002, and 2006; two roots were exposed in 1990, 1995, 1999, and 2003; three roots were exposed in 1991 and 2004; and four roots were exposed in 1994, representing 43.9% of the total roots in the trail.

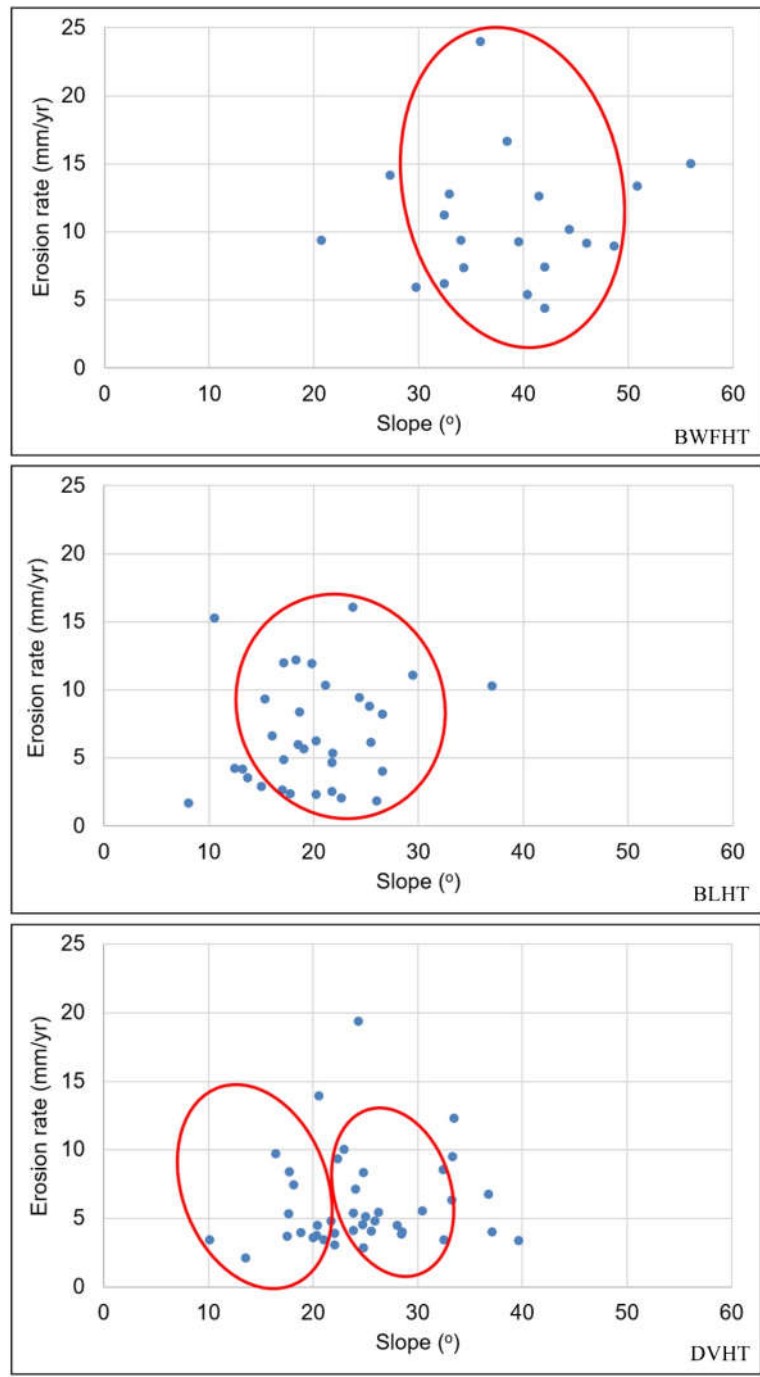

**Figure 7.** Diagram showing the relationship between the slope and erosion along the three tourist trails. Ellipses group all slope angles between 15° and 30° and between 30° and 40°, respectively.

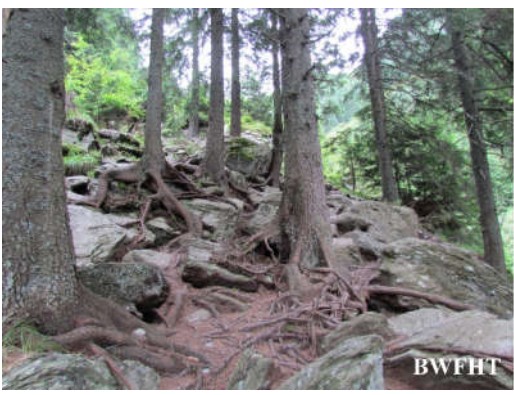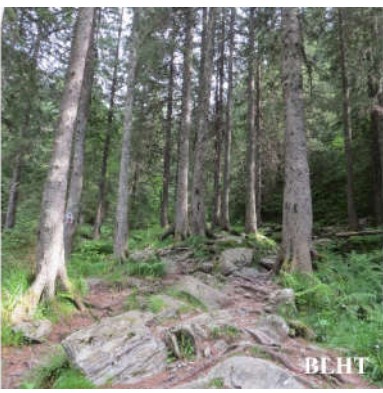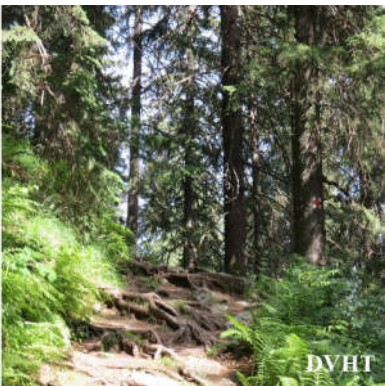

**Figure 8.** Presence of rocks in the three analysed tourist trails.

From another perspective, the years of root exposure can be correlated with the 24 h rainfall data. As mentioned by Refs. [14,99], this correlation can be influenced by the location of the weather station and the localised manifestation of storm cells. The Bâlea weather station is located at an altitude of 2047 m, while the samples were collected at an altitude approximately 500–550 m lower and at a distance of approximately 2.5–3 km. Ref. [107] mentioned that the values of precipitation in 24 h in the central Pyrenees were between 150 and 200 mm, excessively influenced by the Mediterranean. These values play an important role in soil erosion processes. Furthermore, Refs. [13,14] studied the precipitation in the Gorce Mountains in Poland, whose values were more than 70 mm in 24 h, and played a role in soil erosion. Ref. [103] studied the precipitation in the Carpathians and the Sudetes in Central Europe, whose values were between 20–100 mm in 24 h and >100 mm in 24 h ranges, which highlighted its role in erosion processes.

We identified two possible scenarios: the dependence of root exposure and erosion on the amount of precipitation in 24 h (Figure 9b). In the first scenario, a single case of precipitation of more than 100 mm in 24 h (195.6 mm), namely, the most severe heavy rainfall event in the analysed period [108] was recorded on 3 July 1988 with one root being exposed in DVHT (erosion rate of 3.94 mm·y$^{-1}$); in BWFHT, one precipitation event of 107.8 mm in 24 h on 12 July 2009 resulted in the exposure of two roots (erosion rates of 13.42 and 14.21 mm·y$^{-1}$, respectively); one precipitation event of 138.9 mm in 24 h on 11 June 2011 resulted in the exposure of one root in BWFHT (16.7 mm·y$^{-1}$) and DVHT (9.41 mm·y$^{-1}$), respectively. These three years were rainy, with annual rainfall values exceeding the multiyear average. In the second scenario, with rainfall values above 70 mm in 24 h, we identified a few cases of root exposure. On 12 July 2005, with 85.8 mm of precipitation in 24 h, one and two cases of root exposure occurred in BLHT (erosion rate of 8.27 mm·y$^{-1}$) and DVHT (9.52 and 6.81 mm·y$^{-1}$, respectively), respectively. On 1 September 2006, with 78.8 mm of precipitation in 24 h, one, two, and one cases of root exposure occurred in BWFHT (9.42 mm·y$^{-1}$), BLHT (8.40 and 6.67 mm·y$^{-1}$, respectively), and DVHT (5.18 mm·y$^{-1}$), respectively. On 5 September 2007, with 75.2 mm of precipitation in 24 h, we recorded one case of root exposure occurred in BWFHT (11.25 mm·y$^{-1}$), BLHT (9.34 mm·y$^{-1}$), and DVHT (8.43 mm·y$^{-1}$), respectively. On 16 July 2008, with 71.4 mm of precipitation in 24 h, three cases of root exposure were recorded only in BWFHT, with erosion values of 12.8, 12.6, and 9.3 mm·y$^{-1}$, respectively. The majority of cases (71.4%) were recorded in June and July, the rainiest months of the year, with values >160 and >170 mm, respectively. Notably, the 24 h rainfall was recorded in years with high annual rainfall amounts, which exceeded the multiannual average of 1366.2 mm, except in 2006 (1309.5 mm). As in other studies [67,109], and in our case, measurements of the slope of each sampled root did not reveal any connection between slopes and the amount of precipitation falling within 24 h in a short time interval. We consider that although the total number of exposed roots mentioned above is low (19.1% of the total sampled roots), it cannot be accidental, as it is related to the 24 h precipitation.

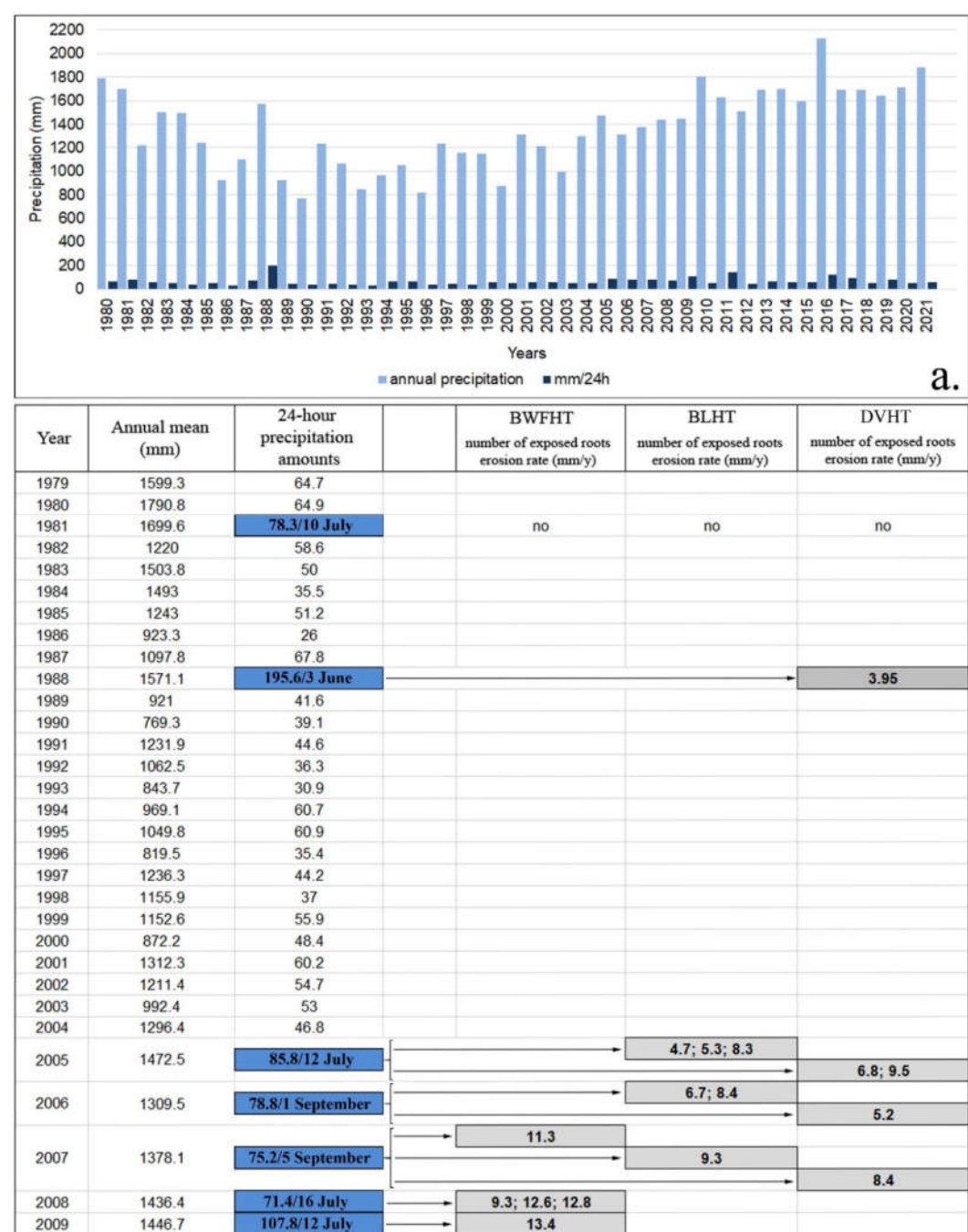

**Figure 9.** Variation of the rainfall regime between 1979 and 2021 in the area of the three analysed trails (**a**) and the impact of the 24 h rainfall on root exposure and erosion (**b**). Blue cells represent precipitation whose values were more than 70 mm in 24 h.

In our study, we encountered some limitations: (a) lack of a uniform spatial distribution of exposed roots suitable for sampling. This was either because of the storm that affected BWFHT by uprooting trees or the occurrence of path stabilisation by railings and wooden supports in the case of BLHT, roots being cut or covered and thus removed from the sampling process [53], or by the presence of large boulders entirely covering the path and fragmenting the presence of exposed roots in certain segments, as in the case of DVHT (Figure 10); (b) difficulties in dating scars in older trees and roots [19]; and (c) validation of erosion rates with the actual flows of hikers entering the trails.

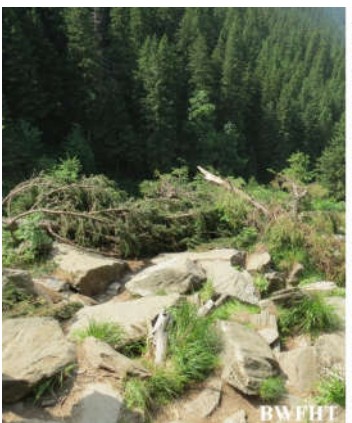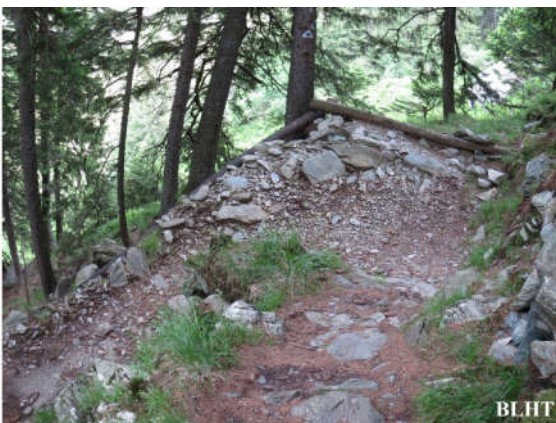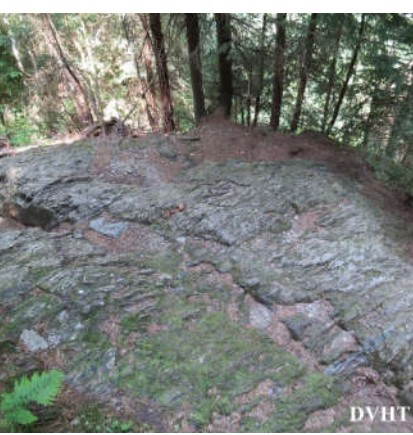

**Figure 10.** 2020 storm (BWFHT), wooden cairns and trail levelling (BLHT), and large boulders covering the entire width of the trail (DVHT).

## 5. Conclusions

The construction of trails on forested mountain slopes in the Făgăraș Mountains for hiking activity resulted in accelerated erosion owing to the exposed ground surface. Thus, the risk of erosion of the slopes of the trails increased due to the removal of vegetation and soil compaction, which reduced the process of water infiltration into the soil.

Our study highlights that mountainous areas are sensitive not only to climatic and environmental impacts [110–112] but also to anthropogenic impacts, represented by hiking activity [5,34,67,68].

The dendrogeomorphological approach is proven to be a valuable tool in assessing erosion rates along tourist trails in coniferous and spruce forests for characterising the influence of human and other activities on the mountain environment in both retrospective and anticipatory ways. Tree roots have great potential for quantitatively assessing the impact of hiking activity in forested areas [33].

For this purpose, we used dendrogeomorphological and anatomical indicators for *P. abies* over 56, 37, and 30 years for BLHT, DVHT, and BWFHT, respectively. The roots sampled from the trails showed clear signs of reduced growth. Therefore, trampling is a disturbing factor for roots and the forest environment [1,6,34,50]. In support of this claim is the number of scars, abundant and synchronous with TRDs in all three trails, especially between the time interval of 1995–2021, when tourists were simultaneous in the BLHT and DVHT. Although the number of scars and erosion rates do not help estimate the number of disturbing elements, such as hikers, the data extracted from the analysis of root rings highlight the magnitude of the impact and enable future predictions [5]. Additionally, in a wet area, the occurrence of quantitatively remarkable rainfall events may contribute to root exposure and accelerate erosion processes in some years but is not a determining factor.

We consider that the results obtained along the three trails under similar environmental conditions, topography, geology, soil, climate, and vegetation are valid only for our study area and cannot be generalised to the scale of the Făgăraș Mountains. However, the numbers of scars and erosion rates were significant, showing the strong impact of tourist activity, along with some natural processes in some years.

**Author Contributions:** Conceptualization, M.V.; Data acquisition, writing original draft, Reviewing and Editing, M.J. and M.V. All authors have read and agreed to the published version of the manuscript.

**Funding:** This research received no external funding.

**Data Availability Statement:** The data presented in this study are available on request from the corresponding author.

**Acknowledgments:** The authors are grateful to Ionel Popa and Patrik Chiroiu for their helpful comments throughout the study and Renata Feher for their support in fieldwork. We would also like to extend our thanks to the anonymous reviewers for their constructive comments as well as the other members of the editorial board of the journal.

**Conflicts of Interest:** The authors declare no conflict of interest.

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
