# Peer review of "Assessment of the Annual Erosion Rate along Three Hiking Trails in the Făgăraș Mountains, Romanian Carpathians, Using Dendrogeomorphological Approaches of Exposed Roots"

_forests, doi:10.3390/f13121993_

Round 1
Reviewer 1 Report
Dear authors,
According to my opinion, your paper is generally excellent with this relatively new approach of erosion rate estimation/calculation. If possible, just improve the main map in Fig. 1 (you can interpolate used - probably 30-m DEM - to better resolution /if not available 5-m DEM for that area; or use shaded base layer). Also, do it is possible/useful to compare your results with the previously available erosion map/model of this area (if any)? Where are the sites bellow with the increased deposition of the eroded material? Thank you.
Reviewer 2 Report
Comments and Suggestions for Authors
This is the review of the manuscript
Journal: MDPI Forests
Submitted to section: Natural Hazards and Risk Management
Manuscript ID: forests-2003664
Authors: Mihai Jula, Mircea Voiculescu
Title: Assessment of the annual erosion rate along three hiking trails in the Făgăraș Mountains, Romanian Carpathians, using dendrogeomorphological approaches of exposed roots
The authors describe the impact of treading paths / trails in the mountains on the erosion rate of these paths. The research uses dendrogeomorphological methods.
The article is very interesting due to the very current problem (not only observed in Fagaras Mountains).
I have few comments and suggestions to authors.
Below I list specific comments:
line 41 - explain what TRD means
line 110 - cold temperatures? rather low temperatures
line 230 - remove from the end of the line (.
line 268 - Corona et al. - add the year of publication
Table 2 - in the caption explain what Ex and Era mean
line 294-295 - that's a sentence for discussion
figure 3 - mark DVHT with a darker line, now the graph is not well visible
figure 4 - for VDHT why is one ellipse for one year, maybe it is better to block the years 19-27 and then it will also be high erosion rate
line 348-349 - this information should appear in the discussion
358-359 - for discussion
lines 363-370 - describe the results only without describing the reasons, the reasons in the discussion
figure 6 - change Nr to No. everywhere, explain in the caption what Ia means
line 383 - min. whole word
figure 9 - in meteorology we do not define mean annual precipitation only amounts / sum; no explanation of what the colors used for part b mean; in b. there should be two values ​​of the number of exposed roots and erosion rate (mm / y) and in the table there is often only 1 number for the case for the position, I do not understand how to read it
Data availability statement: it is worth including the data in an open data repository with the doi number
Reviewer 3 Report
In the manuscript “Assessment of the annual erosion rate along three hiking trails in the FăgăraÈ™ Mountains, Romanian Carpathians, using dendrogeomorphological approaches of exposed roots” Jula and Voiculescu analyze root growth and built a chronology of scars and traumatic resin ducts for each of the three considered trail paths in a Romanian mountain range. The Authors try to relate their chronologies and the erosion rate obtained by them to the usage of each path taking in account the geomorphological, climatical, historical, and social contexts. The authors conclude that the erosion rate measured for each path is probably linked to its use by hikers and thus even if their results are representative of their site, they are hardly relatable to other geographical contexts.
In my opinion, the manuscript is in line with the aims of the journal Forests and its Special Issue Data Acquisition, Methods and Techniques Applied in Sustainable Forest Management and Hazard Mapping in section Natural Hazards and Risk Management reporting original chronologies of scars and traumatic resin ducts that are representative of an accelerated erosion rate on three paths in a mountain range due to human activity. The manuscript accounts an interesting topic on which authors already published but, in my opinion, lacks clarity in some parts and some sections should be improved. Major and minor comments are reported in the attached file, line number is referred to the line reported on the document that can be downloaded from the Journal.

Author Response
Please see the attachement

Round 2
Reviewer 3 Report
In the manuscript “Assessment of the annual erosion rate along three hiking trails in the FăgăraÈ™ Mountains, Romanian Carpathians, using dendrogeomorphological approaches of exposed roots” Jula and Voiculescu analyze root growth and built a chronology of scars and traumatic resin ducts for each of the three considered trail paths in a Romanian mountain range. The Authors try to relate their chronologies and the erosion rate obtained by them to the usage of each path taking in account the geomorphological, climatical, historical, and social contexts. The authors conclude that the erosion rate measured for each path is probably linked to its use by hikers and thus even if their results are representative of their site, they are hardly relatable to other geographical contexts. In my opinion, the manuscript is in line with the aims of the journal Forests and its Special Issue Data Acquisition, Methods and Techniques Applied in Sustainable Forest Management and Hazard Mapping in section Natural Hazards and Risk Management reporting original chronologies of scars and traumatic resin ducts that are representative of an accelerated erosion rate on three paths in a mountain range due to human activity. In the amended version the authors improve the clarity of the text and modify it to reply to most of the comments. However, few minor comments still exist and are reported in the attached file. The line number is referred to the line reported on the amended document that can be downloaded from the Journal. Grey color identify comment to the previous version, red identify the authors reply and black the comment to the amended version.
